# Spatio-Temporal Measurement and Driving Factor Analysis of Ecosystem Service Trade-Offs and Synergy in the Kaidu–Kongque River Basin, Xinjiang, China

**Yujiang Yan [1], Jiangui Li [2,\*], Junli Li [3,\*] and Teng Jiang [4]**

1   College of Economics and Management, Xinjiang Agricultural University, Urumqi 830052, China; yanyujiang1987@163.com
2   College of Forestry and Landscape Architecture, Xinjiang Agricultural University, Urumqi 830052, China
3   State Key Laboratory of Desert and Oasis Ecology, Xinjiang Institute of Ecology and Geography, Chinese Academy of Sciences, Urumqi 830011, China
4   Xinjiang Academy of Forestry Sciences, Urumqi 830018, China
\*   Correspondence: yyj@xjau.edu.cn (J.L.); xjyyj2008@163.com (J.L.)

**Abstract:** The arid ecosystem of the endorheic basin is a complex, integrated ecosystem with diverse functions and significant regional differences. However, measuring the trade-off relationships and external driving mechanisms of the ecosystem services in arid inland basins has always been a challenging task in terms of geography, ecology, and economics. In this study, we utilized meteorological data, land-use and land-cover data, and vegetation NDVI to estimate the five ecosystem services, namely, food supply, water yield, carbon sequestration, habitat quality, and windbreak and sand-fixation supply services, using the RWEQ and InVEST models in the Kaidu–Kongque River Basin, Xinjiang. Bivariate spatial local autocorrelation analysis was employed to measure the trade-off/synergy relationships between these ecosystem services, and GeoDetector was used to identify the impact of the natural environment and human activities on the trade-off relationships between ecosystem services. The results show the following: (1) In the past three decades, all five ecosystem services in the Kaidu–Kongque River Basin increased, with the food supply service being the largest (66.37%), followed by the windbreak and sand-fixation service, with a continuous upward trend of 51.84%. (2) The spatial distribution and pattern changes of each ES exhibit notable spatial heterogeneity, with high-value areas for food supply and carbon-sequestration services situated in the basin's middle reaches with high vegetation cover. Meanwhile, high-value areas for water yield, habitat quality, and windbreak and sand-fixation services are located in Hejing County, upstream of the basin. (3) The trade-offs and synergistic relationships between ecosystem services were explored, with most showing significant correlations at the 0.01 level, and synergistic relationships were predominantly found. (4) The contribution of each ecosystem service was primarily attributable to natural factors rather than human factors. Furthermore, land use/cover type emerged as the dominant factor for spatial differentiation in the integrated ecosystem services of the watershed, followed by elevation and rainfall. By elucidating the trade-offs, spatial heterogeneity, and formation mechanisms of ecosystem services, this study provides a scientific basis for regional ecological planning. Additionally, the study holds practical significance for devising "win-win" policies for regional economic development and ecological balance.

**Keywords:** ecosystem services; trade-off; drivers; Kaidu–Kongque River Basin

## 1. Introduction

Arid ecosystems are essential for providing a diverse range of ecosystem services (ESs) to local populations. However, long-term, unreasonable human exploitation has led to serious environmental problems, severely threatening the ecosystems' health [1]. While increased land productivity, improved oasis microecology, and increased resource

capacity have had positive impacts, they have also led to ecological problems, such as salinization, desertification, and water resource deterioration. The rapid economic and tourism-related development in Northwestern China has further accelerated population growth, industrialization, and urbanization, resulting in challenges for the management and exploitation of water resources in arid endorheic basins. These activities have significantly impacted the ecological health of the region. Therefore, analyzing the trade-offs/synergistic relationships and the factors that influence ESs in arid zones is critical for the effective management of these fragile ecosystems.

ESs play a crucial role in meeting human needs, serving as a vital bridge between the natural environment and societal demands. However, these services are diverse, spatially and temporally variable, and subject to varying degrees of human exploitation and management. Understanding the interrelationships between ESs, their spatial and temporal patterns, and the underlying driving forces is essential for promoting the sustainable management of regional ecosystems, guiding the rational use of natural resources, and enhancing human well-being [2]. Achieving this goal requires identifying and analyzing trade-offs and synergies between ESs [3], avoiding the unintended consequences of enhancing one service to the detriment of others, and facilitating the optimal utilization of multiple services. This approach is critical to translating theoretical research into practical ecosystem management and decision making [4].

In recent years, the interplay of the trade-offs/synergistic relationships between ESs has gained considerable attention from researchers in various fields worldwide, including geography, ecology, and management [5,6]. Hou et al. found that the trade-off relationship between water yield and evapotranspiration weakened over time on the Loess Plateau [7]. Han et al. found an increasing trend in the synergistic relationship between net primary productivity, water yield, and soil conservation in the Sanjiangyuan area [8]. These studies emphasize that trade-offs/synergistic relationships are not only time-dependent, but also scale-dependent. Similarly, Lorilla reported that there was no significant correlation between food production and regulation services at the patch level in the Pedra River, but significant positive correlations were observed when the analysis was expanded to the urban and landscape-level scales [9,10]. Gong et al. also demonstrated the changing status and trade-offs/synergies of ESs in the Bailong River Basin under different scenarios using ES change indexes and trade-offs [11]. In this study, we employed the GeoDetector tool to examine the external drivers of watershed-scale ES trade-off relationships [12,13].

The Kaidu–Kongque River Basin, located in the arid zone of Northwestern China, is a main branch of the Tarim River Basin. Due to its fragile ecological environment, exacerbated by the overexploitation of natural resources, the basin faces significant ecological challenges, such as water quality degradation, water scarcity, and soil erosion. Although previous studies have analyzed the spatial distribution and changes in ESs using long-term time series data, few of them have attached importance to the complex trade-offs/synergies among these services.

In this study, we aimed to investigate the trade-offs, synergistic effects, and influencing factors of typical ESs. We focused on five important ESs, namely: food supply, water yield, carbon sequestration, habitat quality, and windbreak and sand fixation in the Kaidu–Kongque River Basin from 1990–2020. The InVEST model and local indicators of spatial association (LISA) were employed to investigate the spatial distribution of these ESs and to measure their trade-offs and synergies. Additionally, geographic probes were utilized to clarify the driving mechanisms of the trade-offs and synergies. The objectives of this study were to assess the spatial distribution of the selected ESs, measure their trade-offs and synergies, identify trade-off patterns, and analyze the factors influencing these ESs. The present study provides a scientific basis and empirical analysis for ecological and environmental planning and decision making in arid drainage basins.

## 2. Materials and Methods

### 2.1. Study Area

The Kaidu–Kongque River Basin, comprising the Kaidu River, Bosten Lake, and Kongque River, is located in the northeastern part of the Tarim Basin and on the northeastern edge of the Taklamakan Desert (39.54°–43.37° N, 82.91°–90.57° E), covering a total area of 93,532.60 km² (Figure 1). It consists of diverse ecosystems, including mountains, oases, lakes, and deserts. The topography of the basin is characterized by a high altitude in the north and west and a low altitude in the south and east, with ground elevations ranging from 637 to 4817 m. It has a continental arid climate, with significant variations among the upper, middle, and lower reaches of the basin. The multi-year average temperature is −4.3 °C in the upstream mountainous areas, while it ranges from 6 to 12 °C in the midstream and downstream plain areas, with more rainfall in the upper and middle reaches. Annual precipitation ranges from 300 to 500 mm, and it is mainly concentrated in June to August. The soil types, from upstream to downstream, are primarily ice marsh soil, meadow soil, brown desert soil, tidal soil, oasis gray soil, and salt soil, respectively, and the vegetation types are subalpine coniferous forest, alpine shrub forest, river valley forest, deciduous broadleaf, and desert scrub [14]. The study area experienced rapid growth in terms of agricultural and industrial development, serving as an important agricultural production base and a crucial ecological barrier in Northwestern China.

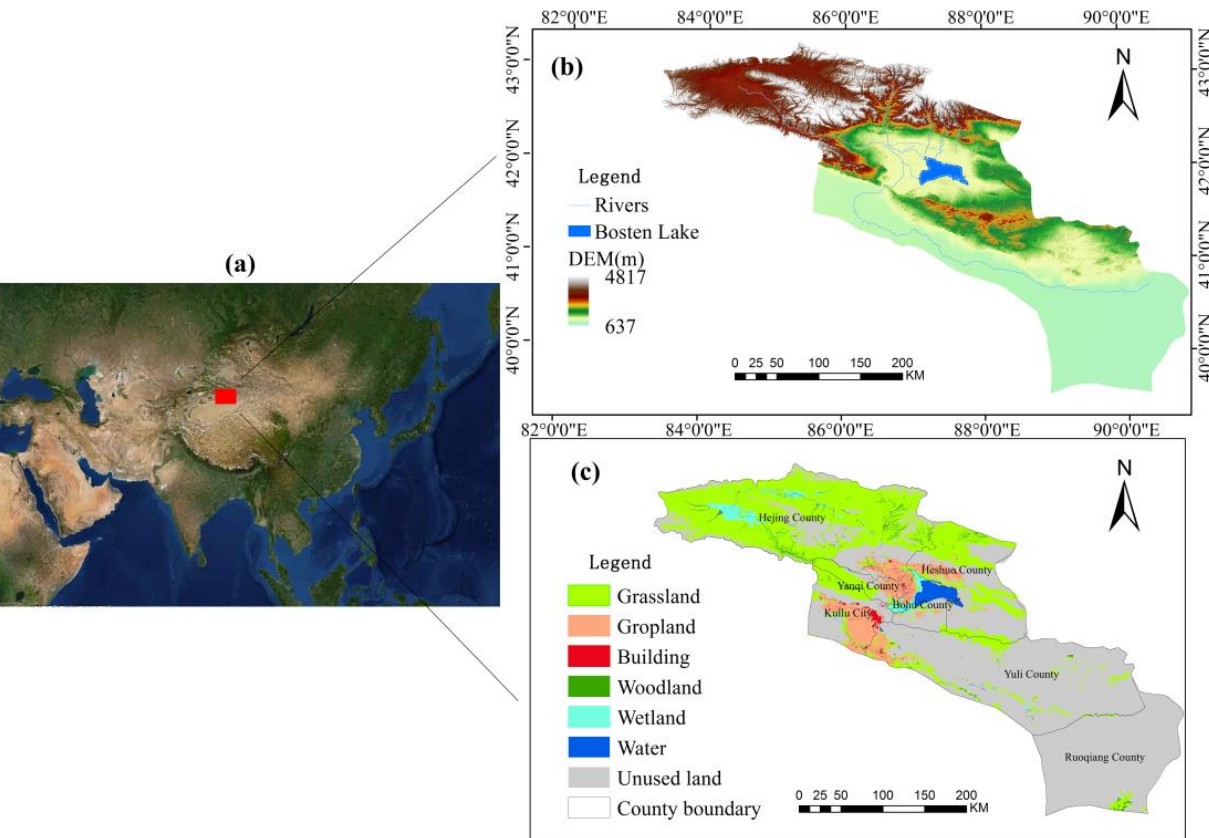

**Figure 1.** Geographic locations of the study area: (**a**) Location of the Kaidu–Kongque River Basin in China; (**b**) terrain of the study; (**c**) land use/cover of the study area.

### 2.2. Data Sources and Processing

The data used in this study (Table 1) were meteorological data (precipitation, air temperature, and solar radiation), soil data (soil type, soil capacity, soil moisture, and groundwater level), remote sensing data, land-use/land-cover change data, basic geographic information, socioeconomic data (such as population density, GDP, and statistical agricultural data), and hydrological data. All of these data sources were resampled to

raster data with a spatial resolution of 1 km × 1 km, and the projection coordinates system was made uniform with UTM projections (Zone 45) to ensure the data's continuity and accuracy.

**Table 1.** Data sources for ES models.

| ES Models | Model Parameters | Data Source | Source |
|---|---|---|---|
| Food supply | Food production (grain, meat, fruit, and fish) Normalized difference vegetation index (NDVI) | Local statistical yearbooks for 1990, 2000, 2010, and 2020 Data Sharing Service System, Institute of Geographical Sciences and Resources, Chinese Academy of Sciences | https://data.casearth.cn/ accessed on 4 July 2021. |
| Water yield | Meteorological data (precipitation, air temperature, and actual water pressure) DEM, soil type, and vegetation root-system data | National Weather Science Data Center Geospatial Data Cloud A Chinese soil dataset based on the World Soil Database, Cold and Arid Regions Science and Data Center | http://data.cma.cn/ accessed on 11 March 2021. http://www.gscloud.cn/ accessed on 10 February 2021. www.geodata.cn accessed on 20 January 2021. |
| Carbon sequestration | Soil depth, soil type, soil capacity, organic carbon, soil thickness, and percentage of gravel | A Chinese soil dataset based on the World Soil Database, Cold and Arid Regions Science and Data Center | www.geodata.cn accessed on 20 January 2021. |
| Habitat quality | Land-use/land-cover changes (1990–2020) | Geospatial Data Cloud Platform | http://www.gscloud.cn/ accessed on 10 February 2021. |
| Windbreak and sand fixation | Wind speed, precipitation, air temperature, sunshine time, and snow cover | National Meteorological Science Data Center National Center for Permafrost and Desert Science Data | http://data.cma.cn/ accessed on 11 March 2021. http://www.ncdc.ac.cn/ accessed on 19 May 2021. |
| Other data | GDP raster data Population raster data Agriculture and other related data | Resource and Environmental Science and Data Center, Chinese Academy of Sciences WordPop Census data for 1990, 2000, 2010, and 2020 Local statistical yearbooks for 1990–2020 | http://www.resdc.cn/DOI accessed on 1 December 2020. https://www.worldpop.org/ accessed on 15 December 2020. |

### 2.3. The Calculation Methods of ES Supplies

According to the classification scheme for ESs proposed by the UN Millennium Ecosystem Assessment (MA, 2005) [15], five types of ESs, including food supply, water yield, carbon sequestration, habitat quality, and windbreak and sand fixation, were selected in order to calculate the ES supplies of the Kaidu–Kongque River Basin based on importance, comprehensiveness, and data availability. The calculation methods are shown in Table 2.

**Table 2.** The calculation methods for ES supplies.

| Type of ES | Calculation Method | | | Reference |
|---|---|---|---|---|
| | **Calculation Method** | | **The Meaning of Each Parameter** | |
| Food supply | $G_i = G_{sum} \times \frac{NDVI_i}{NDVI_{sum}}$ | (1) | where $G_i$ is the yield of grain, meat, fruit, and aquatic products allocated by the i grid; $G_{sum}$ is the total grain, meat, fruit, and aquatic product output in the study area; $NDVI_i$ is the normalized vegetation index of grid i; and $NDVI_{sum}$ is the sum of the NDVI values in the study area. | [2,16] |

**Table 2.** *Cont.*

| Type of ES | Calculation Method | | Reference |
|---|---|---|---|
| | Calculation Method | The Meaning of Each Parameter | |
| Water yield | $WY_x = 1 - \frac{AET_x}{P_x} \times P_x$   (2) | where $WY_x$ denotes the annual water supply service on the raster cell; $AET_x$ denotes the average annual evapotranspiration on the raster cell; and $P_x$ denotes the average annual precipitation on the raster cell. | [17,18] |
| Carbon sequestration | $C_{tot} = C_{above} + C_{below} + C_{soil} + C_{dead}$   (3) | where $C_{tot}$ is the total carbon stock (t·hm$^{-2}$); $C_{above}$ is the aboveground biogenic carbon (t·hm$^{-2}$); $C_{below}$ is the belowground biogenic carbon (t·hm$^{-2}$); $C_{soil}$ is the soil organic carbon (t·hm$^{-2}$); and $C_{dead}$ is the dead organic matter (t·hm$^{-2}$). The carbon density data of the carbon pool table required for the model were mainly obtained from the relevant literature. | [19,20] |
| Habitat quality | $Q_{xj} = H_j \left[ 1 - \left( \frac{D_{xj}^2}{D_{xj}^2 + k^2} \right) \right]$   (4) <br><br> $D_{xj} = \sum_1^r \sum_1^y \left( \frac{w_r}{\sum_{r=1}^n w_r} \right) r_y i_{rxy} \beta_x S_{jr}$   (5) | where $D_{xj}$ is the degree of habitat degradation, ranging from 0 to 1, with higher values representing higher degrees of habitat degradation; $r$ is the threat factor; $y$ is the number of grids corresponding to the threat factor $r$; $Wr$ is the weight of the threat factor; $ry$ is the stress value of the threat factor; $\beta_x$ is the level of habitat protection; $S_{jr}$ is the sensitivity of habitat $j$ to the threat factor $r$; and $i_{rxy}$ is the influence of the threat factor $r$ in grids $y$ on grids $x$. | [21] |
| Windbreak and sand fixation | $SR = \frac{2Z}{sp^2} \times Q_{pmax} \times e^{-\frac{z}{sp}^2} - \frac{2Z}{Sr^2} \times$ <br> $Q_{rmax} \times e^{-\left( \frac{z}{Sr} \right)^2}$   (6) | where $SR$ is sand fixation (t·hm$^{-2}$); $Q_{pmax}$ is the maximum sand transport capacity of potential wind (kg/m); $sp$ is the potential critical plot length (m); and $z$ denotes the calculated downwind distance (m). For this calculation, 50 m was taken; $Q_{rmax}$ is the maximum sand transport capacity of the actual wind (kg/m); and $sr$ is the actual critical plot length (m). | [22,23] |

*2.4. Trade-Off and Synoptic Measurements*

Various analytical methods have been used to interpret the trade-offs and synergies between ESs, including correlation analysis, Ref. [24] cluster analysis, Ref. [25], and PPF curves [26]. In this study, the bivariate local indicators of the spatial association (LISA) model were used to calculate the local Moran's I to reveal the relationships involved in the trade-off system and spatial patterns of ESs. Moran's I is an effective tool for identifying the degree of clustering and spatial patterns of trade-offs/synergies. The Moran's I was calculated as follows:

$$LISA_i = \frac{1}{n} \frac{(x_i - \bar{x})}{\sum_i (x_i - \bar{x})^2} \sum_j w_{ij}(x_i - \bar{x})$$ (7)

where $LISA_i$ is the bivariate local spatial autocorrelation index, and its values range from −1 to 1. Positive and negative values indicate positive and negative spatial correlations, respectively, and the values indicate the degrees of correlations; $w_{ij}$ is the spatial weight matrix between cell $i$ and cell $j$; $x_i$ is the attribute value of cell $i$; $\bar{x}$ is the average of all attributes' values; and $n$ is the total number of regional cells.

*2.5. Driving Factor Analysis*

In arid regions, the functions and value changes of ecosystem services are usually correlated with a series of natural and anthropogenic factors, such as meteorological factors, geomorphologic factors, soil factors, hydrological factors, socioeconomic factors, etc. [26–28] Therefore, in order to analyze the driving factors of changes in the ES value, a total of 6 natural factors and 7 anthropogenic factors were used for multifactor correlation analysis (see Table 3). Approximately 24,445 random points were generated to extract the ES values and driver parameters at each point. In this study, GeoDetector [9,26], a factor detection model, was utilized to evaluate the importance of each factor in the trade-off relationship among ESs and the degree of interaction influence (Equation (7)).

$$q = 1 - \left[ \sum_{h=1}^{L} \sum_{i=1}^{N_h} \frac{(Y_{hi} - \overline{Y}_h)^2}{\sum_{i=1}^{N} (Y_i - \overline{Y})^2} \right] = 1 - \sum_{h=1}^{L} \frac{N_h \sigma_h^2}{N \sigma^2} = 1 - SSW/SST \quad (8)$$

where $q$ is the detection power value of detection factor A, $q \in [0, 1]$, where a greater $q$ value indicates that factor A has a higher trade-off/synergistic effect on ESs in the study area; $N$ and $Nh$ are the sample sizes of the study area, respectively; $\sigma_h^2$ is the discrete variance of factor A within sample $h$; $L$ is the type of each factor in the study area [29]; $h = 1,2,3,..., L$ is the stratification of factor $X$ or dependent variable $Y$; $N_h$ and $N$ are the number of cells in stratum $h$ and the whole area; $\sigma_h^2$ and $\sigma^2$ are the variance of dependent variable $Y$ in stratum $h$ and the whole area, respectively; $SSW$ is the sum of variance within a stratum; and $SST$ is the total variance of the whole region.

**Table 3.** Detecting factor indicators of ESs.

| Driving Factors | Variable Indicators: (Unit) |
|---|---|
| Natural Factors | $X_1$: DEM(m); $X_2$: Annual average precipitation (mm); $X_3$: Average annual evapotranspiration (mm) $X_4$: NDVI; $X_5$: Average annual temperature (°C); $X_6$: Annual average wind speed (s/m) |
| Anthropogenic Factors | $X_7$: Population density (person/km$^{-2}$); $X_8$: Total GDP (billion RMB); $X_9$: Proportion of primary industry (%); $X_{10}$: Proportion of secondary industry (%); $X_{11}$: Proportion of the tertiary industry (%); $X_{12}$: Land-use extent composite index; $X_{13}$: Land-use type |

## 3. Results

*3.1. Spatio-Temporal Changes in ES Supplies*

The ES supplies related to food supply, water yield, carbon sequestration, habitat quality, and windbreak and sand-fixation services throughout the Kaidu–Kongque River Basin in 1990, 2000, 2010, and 2020 were calculated based on the methods listed in Table 2, and the results are shown in Figure 2. In addition, the total supplies of the five ESs and their changes in the study area are shown in Table 4, and their statistical values, organized according to county-level administrative region, are shown in Figure 3.

Specifically, the food supply ES increased by 66.37%, from $24.85 \times 10^4$ t in 1990 to $73.88 \times 10^4$ t in 2020, and the stage with the maximum change rate appeared during the period of 2000–2010. The location with the maximum change rate was the midstream area of the Kongque River, especially in the newly cultivated land areas of Korla City, Yuli County, Hejing County, and Yanqi County since 2000.

The water yield supply ES steadily increased during the period of 1990–2010, but it decreased slightly during the period of 2010–2020 and generally showed an increasing trend, with a rate of 27.16%, over the past 30 years. The majority of the water resources were sourced from the upstream area of the Kaidu River (Figure 3), and the main areas of change were mainly located in these regions.

The carbon sequestration supply ES showed a temporal pattern similar to that of water yield services, and it increased by 9.18% during the period of 1990–2020; the increase in the rate was greatest during the period of 2000–2010. The areas of remarkable improvement were concentrated in Hejing County, Heshuo County, and Yuli County, where the woodland and grassland coverages increased, indicating a significant achievement of the ecological restoration measures and protection policies.

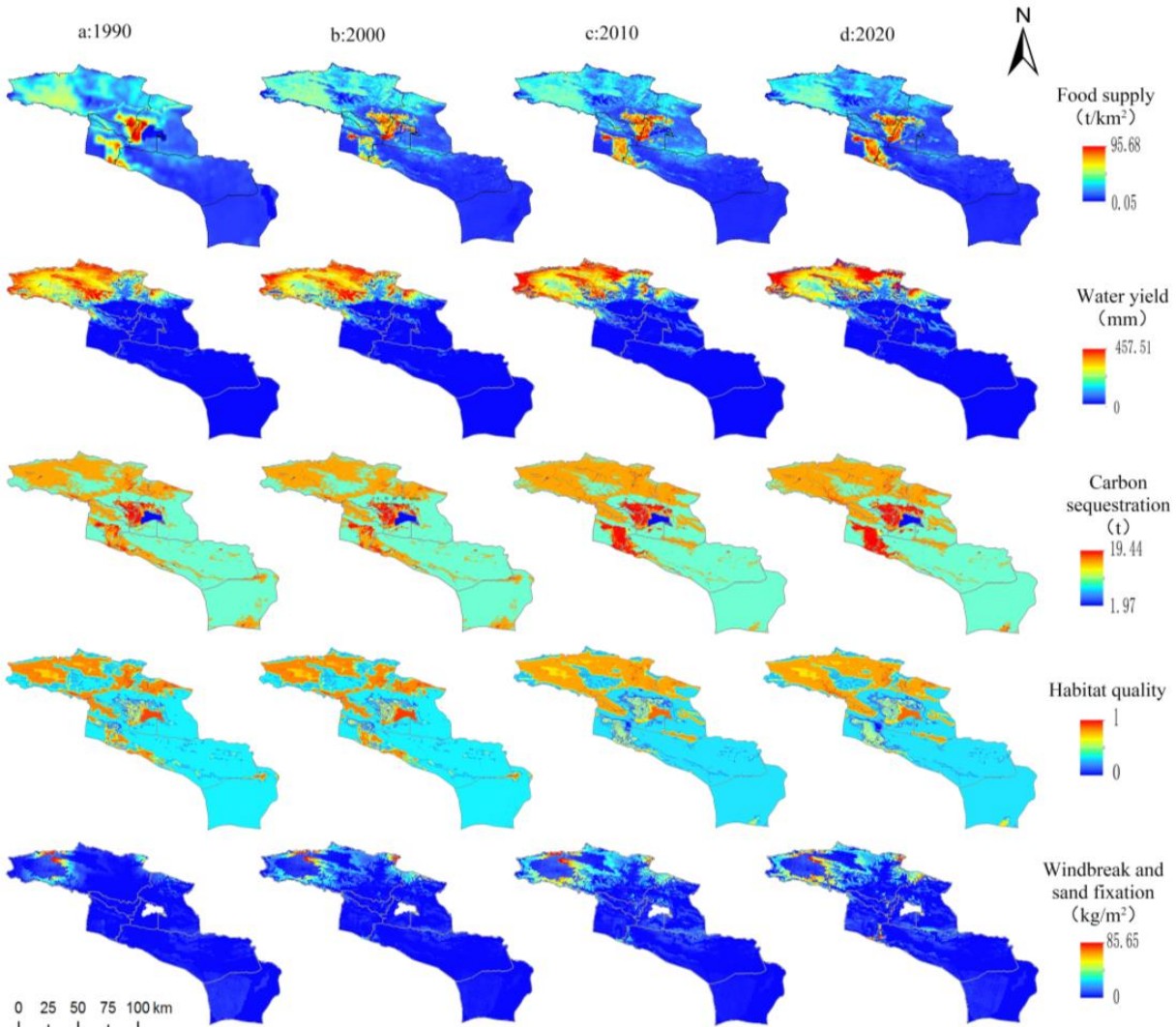

**Figure 2.** Spatio-temporal changes of ES supplies in the Kaidu–Kongque River Basin during the period of 1990–2020.

**Table 4.** Total ES supplies in the study area during the period of 1990–2020.

| Type of ES | 1990 | 2000 | 2010 | 2020 | 1990–2000 | 2000–2010 | 2010–2020 | 1990–2020 |
|---|---|---|---|---|---|---|---|---|
| Food supply (t) | 248,467.07 | 359,468.05 | 692,155.09 | 738,790 | 44.67% | 92.55% | 6.74% | 66.37% |
| Water yield (m³) | $41.27 \times 10^8$ | $44.72 \times 10^8$ | $57.15 \times 10^8$ | $52.48 \times 10^8$ | 8.36% | 27.80% | −8.17% | 27.16% |
| Carbon sequestration ($10^4$ t) | 40,670.39 | 40,501.37 | 44,576.4 | 44,779.4 | −0.42% | 10.06% | 0.46% | 9.18% |
| Habitat quality | 0.435 | 0.434 | 0.46 | 0.459 | −0.23% | 5.99% | −0.22% | 5.23% |
| Windbreak and sand fixation (kg) | $18.17 \times 10^{10}$ | $34.02 \times 10^{10}$ | $93.59 \times 10^{10}$ | $27.59 \times 10^{10}$ | 87.23% | 175.10% | −70.52% | 51.84% |

The habitat quality ES supply experienced a slight increase over the past 30 years, and the areas of improvement in the upstream and midstream linked up to become concentrated, scaled areas (Figure 2), while in the deteriorating areas downstream, especially in Korla City and Yuli County, the habitat quality degraded. Specifically, woodlands and grasslands were prominent in the mountainous regions of Hejing County and Heshuo County, and they have a relatively high habitat quality. However, the levels of habitat quality of Korla City and the Yanqi Basin in the middle reaches of the basin were lesser than those of the upstream due to urbanization, and Yuli County, in the lower reaches, had the worst habitat quality due to sparse vegetation cover.

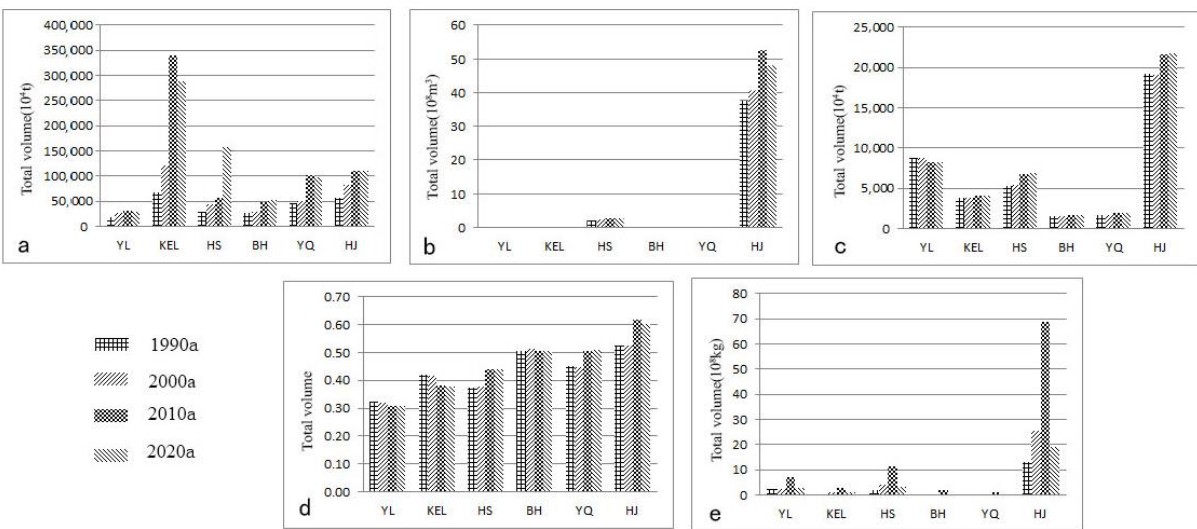

**Figure 3.** Total supplies of ESs and their changes by county and city in the basin from 1990 to 2020: (**a**) food supply; (**b**) water yield; (**c**) carbon sequestration; (**d**) habitat quality; and (**e**) windbreak and sand fixation. Abbreviations: YL: Yuli County; KEL: Korla City; HS: Heshuo County; BH: Bohu County; YQ: Yanqi County; and HJ: Hejing County.

Additionally, the total amount of windbreak and sand-fixation services in the basin showed a trend, first increasing and then decreasing. Specifically, there was a remarkable increase of 175.10% from 2000 to 2010, followed by a decrease of 70.52% from 2010 to 2020. The high-value areas for the windbreak and sand-fixation services were found in the upstream of the Kaidu River in Hejing County, where the potential wind erosion of the soil was greater than the actual wind erosion, leading to a higher supply of wind and sand-control services. Heshuo County and Yuli County were also found to have relatively high windbreak and sand-fixation services. The calculation formula revealed that the magnitude of wind and sand control is dependent on climatic conditions and soil type, with the amount increasing as the vegetation cover increases.

### 3.2. Analysis of Trade-Offs and Synergy Relationships of Watershed ESs

In this study, (Table 5) we found that most ESs in the watershed exhibited a significant correlation at the 0.01 level during the 30-year period. Notably, the Moran index between food supply and water yield was negative, as was the Moran index between food supply and windbreak and sand-fixation services in 1990 and 2010, indicating a significant trade-off between these pairs of ESs. In contrast, the Moran index between the remaining ESs was significantly positive, indicating significant synergy and interactions between them. We observed, through the spatial and temporal measurement distribution of ESs (see Section 3.1), that the high-value area of the food supply was located in the Yanqi Basin and Korla City–Yuli County near the middle reaches of the basin, where a concentration of cultivated land is present. In contrast, the high-value area of the water yield service and windbreak and sand fixation was located in the upper reaches of the basin. The distribution

of high- and low-value areas between these two ESs was opposite, indicating spatial heterogeneity and a trade-off relationship, as was consistent with our earlier findings.

**Table 5.** The Moran's I for the relationships between ESs in the basin from 1990 to 2020.

| ES Pairs | 1990 | 2000 | 2010 | 2020 |
|---|---|---|---|---|
| FD—WY | −0.028 ** | −0.005 | −0.072 ** | −0.052 ** |
| FD—CS | 0.337 ** | 0.362 ** | 0.336 ** | 0.374 ** |
| FD—HQ | 0.164 ** | 0.204 ** | 0.073 ** | 0.122 ** |
| FD—WS | −0.0260 ** | 0.036 ** | −0.008 * | 0.044 ** |
| WY—CS | 0.364 ** | 0.376 ** | 0.482 ** | 0.513 ** |
| WY—HQ | 0.365 ** | 0.366 ** | 0.520 ** | 0.539 ** |
| WY—WS | 0.378 ** | 0.495 ** | 0.592 ** | 0.541 ** |
| CS—HQ | 0.622 ** | 0.599 ** | 0.633 ** | 0.634 ** |
| CS—WS | 0.089 ** | 0.188 ** | 0.369 ** | 0.346 ** |
| HQ—WS | 0.073 ** | 0.126 ** | 0.365 ** | 0.325 ** |

Note: * indicates significance at the 0.05 level, and ** indicates significance at the 0.01 level. Abbreviations: FD: food supply; WY: water yield; CS: carbon sequestration; HQ: habitat quality; and WS: windbreak and sand fixation.

Overall, our findings highlight the importance of understanding the complex interactions between ESs and the spatial heterogeneity of their distribution in order to develop effective management strategies for the sustainable use of natural resources.

### 3.3. ES Driving Factor Detection

The Kaidu–Kongque River Basin's ESs are influenced by a multitude of factors, including natural and anthropogenic factors. To investigate the relative importance of these factors in explaining the spatial variation of ESs, this study utilized GeoDetector. A total of 13 factors were used as factor variables, including population, GDP, precipitation, temperature, land-use type, and land-use intensity. The analysis showed that natural factors, such as land-use type, DEM, precipitation, and NDVI, had a greater contribution to the spatial variation of ESs than did anthropogenic factors, such as GDP and land-use intensity. The factors were ranked according to the magnitude of the mean explanatory power q statistic, with land-use type having the greatest explanatory power, followed by DEM, precipitation, NDVI, GDP, land-use intensity, etc.

The results of this study can provide valuable insights into the main drivers of ES variation in the Kaidu–Kongque River Basin and can inform policymakers and stakeholders about the need to consider both natural and anthropogenic factors in ecosystem management and decision making.

Our results revealed that land-use intensity, GDP, and the proportion of secondary industry value have the highest q values and explanatory power, exceeding 50% for food supply services (Figure 4a). These are primarily driven by human factors. On the other hand, natural factors, such as average evaporation, temperature, precipitation, and DEM, are the major drivers behind water yield services, accounting for more than 50% of the explanatory power (Figure 4b). For carbon sequestration services, land-use type has the highest q value (96.69%), followed by NDVI (47.10%), which is influenced by both natural and human factors (Figure 4c). For habitat quality services, land-use type and DEM have the greatest explanatory power, exceeding 50%, while the average precipitation has an explanatory power of 44.17%, primarily driven by natural factors (Figure 4d). Finally, windbreak and sand-fixation services are mainly influenced by natural factors, including average temperature, DEM, and evaporation, with a level of explanatory power exceeding 30% (Figure 4e). In terms of managing ESs sustainably, it is crucial to reduce human interference while paying close attention to the impact of natural factors. Optimizing the allocation of land resources and constructing a secure pattern can help to achieve a stable supply of ESs, ultimately promoting the optimization of land ecosystem functions.

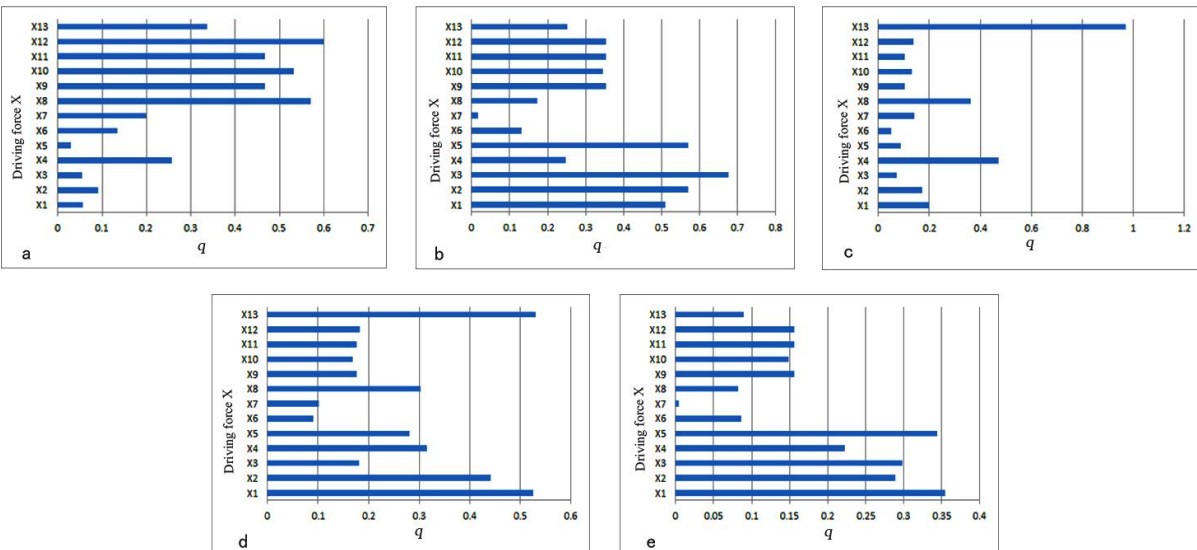

**Figure 4.** Single-factor drivers of watershed ESs: (**a**) food supply; (**b**) water yield; (**c**) carbon sequestration; (**d**) habitat quality; and (**e**) windbreak and sand fixation. Factors: X1: DEM; X2: annual average precipitation; X3: average annual evapotranspiration; X4: NDVI; X5: average annual temperature; X6: annual average wind speed; X7: population density; X8: total GDP; X9: proportion of primary industry; X10: proportion of secondary industry; X11: proportion of tertiary industry; X12: land-use extent composite index; and X13: land-use type.

## 4. Discussion

### 4.1. Assessment of ESs

In this study, we focused on ES changes in the arid zone, where vegetation growth is highly sensitive to hydrothermal conditions, making precipitation a critical factor influencing the supply of ESs. Our analysis revealed that ESs such as food supply, carbon sequestration, habitat quality, and windbreak and sand fixation are generally more abundant in the northwestern mountainous and central regional plain watersheds with higher vegetation cover, and less so in the downstream desert zone areas. As water is the main limiting factor for productivity in arid regions, the increased water yield in the basin over the 30-year study period has contributed to the overall increase in ESs. Additionally, improvements in water conservation techniques and the construction of water conservation facilities have expanded the distribution area of vegetation and improved productivity levels. Our findings underscore the importance of considering the interplay between hydrothermal conditions and ESs in arid zones, and they suggest that strategic investments in water conservation and management could enhance the ES supply and support sustainable ecosystem management in such regions [30,31].

### 4.2. ES Trade-Offs

The trade-offs between ESs are a crucial consideration for sustainable management. In this study, trade-offs mainly occurred between food supply and water yield, and between food supply and windbreak and sand-fixation services, which is consistent with previous studies [32,33]. The spatial heterogeneity of ESs in the arid zone, with water yield regions in the upper reaches and food production in the midstream plains, may explain this phenomenon. Woodlands and grasslands play a positive role in multiple ESs, including food production, water conservation, carbon sequestration, and reduction in wind and sand erosion [34–36]. Therefore, we recommend prioritizing crop cultivation in the central agricultural production areas to meet local food demand, focus on ecological protection and regulating services in the northern and western mountainous areas, and maintain the status quo with sand-fixation measures in the downstream desert areas where there is less human interference. These findings have significant implications for ecosystem management

decisions in arid zones and beyond, highlighting the importance of considering the trade-offs and spatial heterogeneity of ESs [36].

*4.3. ES Impact Factors on ES Changes*

In this study, we explored the magnitude of the explanatory power q-values of 5 ESs and 13 influencing factors using geographic probes, and the results show that the influence of natural factors was greater than that of human factors; for example, the windbreak and sand-fixation services increased with each, especially for the strongest sand-fixation services in 2010. The enhanced areas were mainly in the alluvial fan areas located at the river outlet, and the increases were also more significant in the middle and lower reaches of the watershed. These areas are mainly influenced by meteorological factors, especially wind speed, rainfall, evapotranspiration, and solar radiation on soil wind erosion, and increases in vegetation play an important role in hindering the further spread of wind and sand [37]. Since there are significant synergistic relationships between ESs, it is evident that human activities can influence the interrelationships between ESs and promote their synergistic relationships; for example, increasing arable land can increase food production and habitat quality [38]. Therefore, regional ecological and environmental managers should develop relevant and effective policies to achieve the coordinated development of the regional ecology and economy in the Kaidu–Kongque River Basin.

The dominance of land-use/land-cover change as a key factor affecting the trade-offs and synergistic relationships of ESs has been demonstrated [39,40]. As a complex system of surface elements covered by both natural and artificial structures, the Kaidu–Kongque River Basin is subject to influence by both natural and human factors. To ensure the sustainable output of ESs in the basin, ecological resource protection and management should be further strengthened, with particular attention given to the construction of ecological security in the core area of water-related environmental protection. This should be achieved by establishing a comprehensive ecological civilization construction work system, from the top-level design of the Kaidu–Kongque River Basin to the implementation of regulations on water, ecological, and environmental protection in Bayinguoleng Mongol Autonomous Prefecture; land-use planning and control; industrial structure adjustment; the enforcement of penalties for enterprise pollution discharge; as well as the standardization of environmental law enforcement and supervision. Given the high contribution of natural factors to ESs in the basin, future studies should focus on the impact of extreme climate change on ESs and their trade-off relationships. Establishing spatial and temporal relationships between climate extremes and ESs will enable us to understand the changing relationships between ESs in the context of global change. Such research will provide a theoretical basis for regional ecological environment construction and the development of a harmonious human–Earth relationship.

## 5. Conclusions

It is crucial to understand the supply relationships of ESs for improving regional ecological management and promoting sustainable development. In this study, we aimed to identify the temporal and spatial patterns of ES supply and their interrelationships in the Kaidu–Kongque River Basin in the period from 1990 to 2020. Our findings provide a useful tool for promoting sustainable development in the region. Additionally, we utilized GeoDetector to quantify the spatial heterogeneity of different influencing factors on five ESs, and it has proven to be an effective tool. Our results highlight the complex relationships between ESs and provide important insights for regional ecological management and policy making. Our findings revealed the following.

(1) Over the past three decades, the total supplies of all five ecosystem services in the Kaidu–Kongque River Basin have increased, with notable spatial heterogeneity and patterns of change. Specifically, the food supply and carbon sequestration services have had similar spatial patterns, where high-value areas were concentrated in the middle reaches of the watershed, characterized by high vegetation cover. Meanwhile,

water yield, habitat quality, and windbreak and sand-fixation services had similar spatial patterns, where high-value areas were located at Hejing County in the upstream of the Kaidu River. Over the past 30 years, the habitat quality and windbreak and sand-fixation services have spread to the middle reaches. These findings provide significant implications for promoting sustainable development and effective ecological management in the Kaidu–Kongque River Basin.

(2) The spatial changes in the ecosystem services were influenced by both natural and human factors. The land-use type was the most significant factor in explaining the spatial variation of comprehensive ecosystem services in the Kaidu–Kongque River Basin, followed by elevation and precipitation. Furthermore, the contribution rates of different factors to each type of ecosystem service are distinct, with natural factors generally having a larger impact than human factors. This suggests that the effective management of land use and natural resources is critical for improving the supply and distribution of ecosystem services in the watershed, and this should be a priority for policymakers and stakeholders working towards sustainable development goals.

In this study, five typical ecosystem services were assessed in the Kaidu–Kongque River Basin, and some socioeconomic data derived from statistical yearbooks were consulted to determine the accuracy of the findings. However, there are still several challenges for data acquisition and coordination. These include the inconsistent resolution of remote-sensing data, missing attribute values of vector data, and the unavailability of open key data, such as groundwater data. Furthermore, the selected time period for this study was not very extensive, and the analysis failed to capture the detailed patterns and trends of ESs and their interrelationships, as these services have spatial variability and time-lag effects [41,42]. In the future, we recommend that continuous and long-term time series data be collected in order to conduct a detailed analysis of the spatial and temporal variations in different drivers and methods of ES provisioning. This approach will provide a better understanding of the complex interactions among various factors and their influences on the ESs in the Kaidu–Kongque River Basin.

**Author Contributions:** Conceptualization, methodology, software, writing—original draft preparation, and visualization: Y.Y. and J.L. (Jiangui Li); validation, formal analysis, data curation: J.L. (Junli Li); writing—review and editing: Y.Y.; supervision, project administration, and funding acquisition: T.J. All authors have read and agreed to the published version of the manuscript.

**Funding:** This research was funded by the Tianshan Talent-Science and Technology Innovation Team (2022TSYCTD0006), and the Third Integrated Scientific Expedition Project in Xinjiang (2021xjkk1403).

**Institutional Review Board Statement:** Not applicable.

**Informed Consent Statement:** Not applicable.

**Data Availability Statement:** Not applicable.

**Acknowledgments:** The data that support the findings of this study are available upon request from the corresponding author. The data are not publicly available due to privacy or ethical restrictions.

**Conflicts of Interest:** The authors declare no conflict of interest.

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
