# Peer review of "Spatio-Temporal Measurement and Driving Factor Analysis of Ecosystem Service Trade-Offs and Synergy in the Kaidu–Kongque River Basin, Xinjiang, China"

_sustainability, doi:10.3390/su151612164_

Round 1

Reviewer 1 Report

Dear editor-in-chief

Although the research has its merit and interesting results, some structural problems in the methods presented put it into a situation of untrustful condition. That is why my recommendation is for the paper’s rejection.

There is no clarity on how the ecosystem services were estimated (Table 1). Pixel by pixel? By class? All of them were estimated in the same manner? So, it is impossible to understand and replicate the research.

In the same way, some details on equations provided (Table 1) were not provided or are not clear. Where the data for the ???? index for the Food supply estimation came from? The habitat quality estimation procedure is confusing as the information presented does not correspond to the variables in the equation.

There’s no information on how the Driving forces (line 144) were identified and put into the analysis. It is not precise and not reproducible.

And finally, considering the sentence in lines 180-181 (results), how did authors analyze the spatial temporal change in the ecosystem services provided over the 30 years? No details were presented in the material and methods section.

Author Response

Response to Reviewer 1 Comments

Dear Reviewers,

Thank you for your letter and for the reviewers’ comments concerning our manuscript entitled “Spatiotemporal measurement and driving factors analysis of 2 ecosystem service trade-off and synergy in Kaidu-Kongque 3 River Basin, Xinjiang, China” (ID: 2315910).Those comments are all valuable and very helpful for revising and improving our paper, as well as the important guiding significance to our researches. We have studied comments carefully and have made correction which we hope meet with approval. Revised portion are marked in red in the paper. The main corrections in the paper and the responds to the reviewer’s comments are as flowing:

  1. Comment 1: Although the research has its merit and interesting results, some structural problems in the methods presented put it into a situation of untrustful condition. That is why my recommendation is for the paper’s rejection.

Response: Thanks much for reviewer’s valuable suggestions for our manuscript. We do know that the manuscript need to be improved in many aspects, and we also admit that there are many inappropriate structures and language expressions which made the manuscript uneasy to understand. Therefore, we made major revisions for the paper’s structure and language expressions, so as to make it more readable and easier to understand. It's worth noting that the method in the manuscript was not original(Zhu P, Liu X, Zheng Y H, et al. Tradeoffs and synergies of ecosystem services in key ecological function zones in northern China. Acta Ecologica Sinica, 2020, 40(23): 8694-8706.), but the main purpose of this manuscript was to study the spatial and temporal distribution of the supply of five ecosystem services in the Kaidu-Kongque drainage basin, and to analyze the responses of ecosystem service changes to natural and anthropogenic factors. The data sources are obtained from the open and published water resource bulletin data, climate data and statistical yearbook, and the results had been validated and reliable. We sincerely hope that the review expert can give us a chance, and read the modified manuscripts and provide more constructive suggestions.

  1. Comment 2: There is no clarity on how the ecosystem services were estimated (Table 1). Pixel by pixel? By class? All of them were estimated in the same manner? So, it is impossible to understand and replicate the research.

Response: Thank you very much for your comments. We are sorry that the 2.5 Data source and processing lay behind the 2.2 ES calculations, and it is not easy to understand how the model parameters are derived. So we have changed its orders, 2.2. Data sources and processing, 2.3 ES calculations. And section 2.2 has listed the parameters of ES models and their data sources. All the model parameters are transformed as grid raster layers, and the grid size is 1km×1km. The water yield, carbon sequestration, and habitat quality are calculated by using the InVEST model, and food supply is assigned according to the ratio of raster NDVI values; windbreak and sand fixation is assessed using RWEQ (Revised Wind Erosion Equation). We have made the major revisions in section 2, and all the comments have been accepted, and the modifications can be shown in the Section 2.

  1. Comment 3: In the same way, some details on equations provided (Table 1) were not provided or are not clear. Where the data for the ???? index for the Food supply estimation came from? The habitat quality estimation procedure is confusing as the information presented does not correspond to the variables in the equation.

Response: Thanks for the suggestion. We simplified the calculation method introduction due to space limitations. And we have add more commentary words for each calculation formula, as can be shown in Table 2. The data source of Gsum were aggregated from the statistical yearbook, and the calculation method came from inVEST model[2,16] . And more details have been provided to introduce the habitat quality estimation procedure, as can be shown in Table 2 of the modified manuscript.

Table 2. The calculation methods of ESs.

Types of ES

Calculation method

Reference

Calculating method

The meaning of each parameter

Food supply

(1)

Where  is the yield of grain, meat, fruit and aquatic products allocated by i grid;  is the total grain output, meat, fruit and aquatic products output in the study area.  is the normalized vegetation index of grid i;  is the sum of NDVI values in the study area.

[2,16]

Water yield

 ï¼ˆ2)

Where:  denotes the annual water supply service on the raster cell;  denotes the average annual evapotranspiration on the raster cell;  denotes the average annual precipitation on the raster cell.

[17-18]

Carbon sequestration

 ï¼ˆ3)

where  is total carbon stock (t·hm-2),  is aboveground biogenic carbon (t·hm-2),  is belowground biogenic carbon (t·hm-2),  is soil organic carbon (t·hm-2) and  is dead organic matter (t·hm-2). The carbon density data in the carbon pool table required for the model were mainly referred to relevant literature

[19-20]

Habitat quality

(4/5)

Where  is the degree of habitat degradation, ranging from 0 to 1, with higher values representing higher degrees of habitat degradation;  is the threat factor;  is the number of grids corresponding to the threat factor r;  is the weight of the threat factor;  is the stress value of the threat factor;  is the level of habitat protection;  is the sensitivity of habitat j to the threat factor r;  is the influence of the threat factor r in grids y on grids x

[21]

Windbreak and sand fixation

 ï¼ˆ6)

where  is sand fixation (t·hm-2);  is the maximum sand transport capacity of potential wind (kg/m),  is the potential critical plot length (m), z denotes the calculated downwind distance (m), 50m was taken for this calculation;  is the maximum sand transport capacity of the actual wind (kg/m), sr is the actual critical plot length (m).

[22-23]

  1. Response to comment: There’s no information on how the Driving forces (line 144) were identified and put into the analysis. It is not precise and not reproducible. .

Response: According the reviewer’s suggestion, we made major revisions about the method introduction. The GeoDetector model proposed by Wang Jinfeng[3,26] was used to analyze the driving factors. And this model is widely used in many areas and has been proved effective. We have modified the methods in Section 2.5 Driving factor analysis, as can be shown followed,

In arid regions, the functions and value changes of ecosystem services are usually correlated with a series of natural and anthropogenic factors, such as Meteorological fac-tors, geomorphologic factors, soil factors, hydrological factors, socio-economic factors, etc. Therefore, in order to analyze the driving factors of ES value changes, a total of 6 natural factors and 7 anthropogenic factors were used for multi-factor correlation analysis (Table 3). About 24445 random points were generated to extract the ES values and driver parameters at each point. In this study, the GeoDetector[9,26], a factor detection model, was utilized to evaluate the importance of each factor on the trade-off relationship among ESs and the degree of interaction influence (Equation 7).

  1. Comment 5: And finally, considering the sentence in lines 180-181 (results), how did authors analyze the spatial temporal change in the ecosystem services provided over the 30 years? No details were presented in the material and methods section.

Response: Thanks for the suggestion. Maybe the inappropriate expressions lead to misunderstanding, and we have rewritten the sentence in the lines 180-181. We have added all the materials and methods for the results. As the ES supplies had been mapped and the total ES supplies have been counted, so the ES spatio-temporal changes were analyzed based on these results. And the modified paragraph was listed below:

The ES supplies of food supply, water yield, cabon sequenstration, habitat quality and windbreak and sand fixation services in 1990, 2000, 2010 and 2020 throughout the Kaidu-Kongque river basin were calculated based on the methods in Table 2, and the re-sults showed in Figure 2. Moreover, the total supplies of the five ESs and their changes in the study area were also showed in Table 4, and their statistics by county-level adminis-trative regions were shown in Figure 3.

Specifically, the food supply ES increased by 66.37%, from 24.85×104t in 1990 to 73.88×104t in 2020, and the stage with maximum changing rate appeared during 2000-2010, while the place with the maximum changing rate located in the midstream of Kongque river, especially in the new cultivated land area in Korla City, Yuli County, Hejing County, and Yanqi County since 2000.

The water yield supply ES steadily increased during 1990-2010 but decreased slightly during 2010-2020, and generally showed a increasing trend with a rate of 27.16% in the past 30 years. As the majority of water resources were sourced from the upstream of the Kaidu river (Fig 3), and the mainly changing areas mainly located at these regions.

The carbon sequestration supply ES showed a similar temporal pattern with that of water yield services, and it increased by 9.18% during 1990-2020, especially, the increas-ing rate was largest during 2000-2010. The remarkable improvement areas were concen-trated in Hejing County, Heshuo County, and the Yuli County, where the woodland and grassland coverages increased, indicating a significant achievement for the ecological restoration measures and protection policies.

We look forward to hearing from you regarding our submission. We would be glad to respond to any further questions and comments that you may have”

Reviewer 2 Report

In this paper, five ecosystem services, namely food supply, water yield, carbon sequestration, habitat quality, and windbreak and sand fixation were estimated for the Kaidu-Kongque River Basin, Xinjiang, China. A bivariate spatial local autocorrelation analysis was then employed to measure the trade-off/synergy relationship of these ecosystem services, while geo detector was used to identify the impact of the natural environment and human activities on the trade-off relationship of ecosystem services.

The study is not particularly original, since it uses already well established methods (i.e. InVEST model) to derive information that can have mainly a local relevance for the management of arid environments in China.

However, the paper is well written and in my opinion can be considered for publication.

I only have the following minor concerns:

-          Spatial resolution of the analysis is 1km x 1km, which in my opinion is far too large: authors should discuss the limitations of the spatial scale used, i.e. overlapping between different ecosystems.

-          English language is not bad, but it can be further improved

-          All figure and table legends should be checked since they contain some template sentences (e.s. “this is a figure”);

-          Conclusion is not a summary of the work, it should be shortened focussing on the statement if the experimental hypothesis is verified, the unsolved problems and the future perspectives.

-          Please include some quantitative data in the abstract, e.g. line 23 “with the food supply service experiencing the largest increase (66.37%).

Author Response

Response to Reviewer 2 Comments

Dear Reviewers,

Thank you for your letter and for the reviewers’ comments concerning our manuscript entitled “Spatiotemporal measurement and driving factors analysis of 2 ecosystem service trade-off and synergy in Kaidu-Kongque 3 River Basin, Xinjiang, China” (ID: 2315910).Those comments are all valuable and very helpful for revising and improving our paper, as well as the important guiding significance to our researches. We have studied comments carefully and have made correction which we hope meet with approval. Revised portion are marked in red in the paper. The main corrections in the paper and the responds to the reviewer’s comments are as flowing:

Comment 1. Response to comment: Spatial resolution of the analysis is 1km x 1km, which in my opinion is far too large: authors should discuss the limitations of the spatial scale used, i.e. overlapping between different ecosystems.

Response: Many thanks to the reviewers for their constructive suggestions on our manuscript. The high spatial resolution will provide more details and more precise results. However, beside the remote sensing data, the other raster data used in the manuscript, for example meteorological data, soil data, basic geographic information data, socio-economic data are resampled and rasterized by scatter points or text materials, and the grid size cannot be small due to the data source limitations. Secondly, the calculation of ecosystem services in this manuscript was based on InVEST model, and most published paper about the InVEST model take the grid size of the input data as 1km. Thirdly, there are also similar topic as the manuscript, the grid size of the data source also used 1km. Therefore, in our study area with 9.73 × 104 km2, we think 1km is appropriate.

Comment 2. English language is not bad, but it can be further improved

Response: Thank you for your positive comments on this manuscript, and we have made major revisions for the manuscript, especially for the English expressions and discussions, as can be checked in the revised manuscript.

Comment 3: All figure and table legends should be checked since they contain some template sentences (e.s. “this is a figure”);

Response : We are very sorry for our incorrect expressions, and we have checked all the figures and table legends, and make a professional presentation, as can be checked in the revised manuscript.

Comment 4: Conclusion is not a summary of the work, it should be shortened focussing on the statement if the experimental hypothesis is verified, the unsolved problems and the future perspectives.

Response : According the reviewer’s suggestion, We have made significant changes to our conclusions, as follows:

It is cruicial to understanding the supply relationships of ESs for improving regional ecological management and promoting sustainable development. In this study, we aimed to identify the temporal and spatial patterns of ES supply and their interrelationships in the Kaidu-Kongque River basin from 1990 to 2020. Our findings provided a useful tool for promoting sustainable development in the region. Additionally, we utilized GeoDetector to quantify the spatial heterogeneity of different influencing factors on 5 ESs, and it is proven to be an effective tool. Our results highlight the complex relationships between ESs and provide important insights for regional ecological management and policy-making. Our findings reveal that:

(1) Over the last three decades, the total supplies of all five ecosystem services in the Kaidu-Kongque River basin had increased, with notable spatial heterogeneity and change patterns. Specifically, the food supply and carbon sequestration services had similar spa-tial patterns, where high-value areas were concentrated in the middle reaches of the wa-tershed characterized by high vegetation cover. Meanwhile, the water yield, habitat quality, and windbreak and sand fixation services had similar spatial patterns, where high-value areas were located at Hejing County in the upstream of Kaidu river. Over the past 30 years, the habitat quality and windbreak and sand fixation services had spread to the middle reaches. These findings provide significant implications for promoting sustainable de-velopment and effective ecological management in the Kaidu-Kongque River basin.

(2) The spatial changes of ecosystem services were influenced by both natural and human factors. Land use type was the most significant factor in explaining the spatial variation of comprehensive ecosystem services in the Kaidu-Kongque River basin, fol-lowed by elevation and pricipitation. Furthermore, the contribution rates of different fac-tors to each type of ecosystem service are distinct, with natural factors generally having a larger impact than human factors. This suggests that effective management of land use and natural resources is critical for improving the supply and distribution of ecosystem services in the watershed, and should be a priority for policymakers and stakeholders working towards sustainable development goals.

  1. Please include some quantitative data in the abstract, e.g. line 23 “with the food supply service experiencing the largest increase

Response: We have re-written this part according to the Reviewer’s suggestion, as follows: (1)over the past three decades, the total amount of all five ecosystem services in the Kaidu-Kongque River Basin has increased, with the food supply service experiencing the largest increase (66.37%), followed by the windbreak and sand fixation service, with a continuous upward trend of 51.84%.;

We look forward to hearing from you regarding our submission. We would be glad to respond to any further questions and comments that you may have”

Reviewer 3 Report

Dear authors and editor,

I reviewed the ms. "Spatiotemporal measurement and driving factors analysis of 2 ecosystem service trade-off and synergy in Kaidu-Kongque 3 River Basin, Xinjiang, Chinese". The work is well done, presents very important contributions, but need some efforts to improve the figures and conclusion. I have some suggestions that should be incorporated.

- All tables: Check the title, its not necessary insert "This is a table. Tables should be Quantitative model of ES supply and demand assessment." . I believ that is just some error.

- All figures: The same! Please, check the titles!

- Figures 2, 3 and 4: no clear. Please improve the apresentation because its hard to read.

- Conclusion: the authors must be objective. Conclusion seems to be another discussion. Please, check the objectives and be objective in conclusion.

Author Response

Response to Reviewer 3 Comments

Dear Reviewers,

Thank you for your letter and for the reviewers’ comments concerning our manuscript entitled “Spatiotemporal measurement and driving factors analysis of 2 ecosystem service trade-off and synergy in Kaidu-Kongque 3 River Basin, Xinjiang, China” (ID: 2315910).Those comments are all valuable and very helpful for revising and improving our paper, as well as the important guiding significance to our researches. We have studied comments carefully and have made correction which we hope meet with approval. Revised portion are marked in red in the paper. The main corrections in the paper and the responds to the reviewer’s comments are as flowing:

Comment 1: All tables: Check the title, it’s not necessary insert "This is a table. Tables should be Quantitative model of ES supply and demand assessment" . I believe that is just some error.

Response:  We are very sorry for our incorrect writing expression, we have modified all figures and tables throughout the text with a professional expressions, thank you for your suggestions.

Comment 2: All figures: The same! Please, check the titles!

Response:  Thank you for your suggestions, we have modified all these in the revised manuscript.

Comment 3: Figures 2, 3 and 4: no clear. Please improve the presentation because it’s hard to read.

Response: Thanks much for reviewer’s constructive suggestions about our manuscript, We have modified the Figure 2 to make it easier to understand. Furthermore, the resolution of the histograms of Figs.3 and 4 is appropriately improved, and the other unmodified parts are requested to be informed in detail by the reviewer, and we will further modify and improve them.

Figure 2. Spatio-temporal variation in the supply of ESs the Kaidu-Kongque River Basin from1990 to 2020.

Comment 4: Conclusion: the authors must be objective. Conclusion seems to be another discussion. Please, check the objectives and be objective in conclusion.

Response : According the reviewer’s suggestion, We have made significant changes to our conclusions, as follows:

It is cruicial to understanding the supply relationships of ESs for improving regional ecological management and promoting sustainable development. In this study, we aimed to identify the temporal and spatial patterns of ES supply and their interrelationships in the Kaidu-Kongque River basin from 1990 to 2020. Our findings provided a useful tool for promoting sustainable development in the region. Additionally, we utilized GeoDetector to quantify the spatial heterogeneity of different influencing factors on 5 ESs, and it is proven to be an effective tool. Our results highlight the complex relationships between ESs and provide important insights for regional ecological management and policy-making. Our findings reveal that:

(1) Over the last three decades, the total supplies of all five ecosystem services in the Kaidu-Kongque River basin had increased, with notable spatial heterogeneity and change patterns. Specifically, the food supply and carbon sequestration services had similar spa-tial patterns, where high-value areas were concentrated in the middle reaches of the wa-tershed characterized by high vegetation cover. Meanwhile, the water yield, habitat quality, and windbreak and sand fixation services had similar spatial patterns, where high-value areas were located at Hejing County in the upstream of Kaidu river. Over the past 30 years, the habitat quality and windbreak and sand fixation services had spread to the middle reaches. These findings provide significant implications for promoting sustainable de-velopment and effective ecological management in the Kaidu-Kongque River basin.

(2) The spatial changes of ecosystem services were influenced by both natural and human factors. Land use type was the most significant factor in explaining the spatial variation of comprehensive ecosystem services in the Kaidu-Kongque River basin, fol-lowed by elevation and pricipitation. Furthermore, the contribution rates of different fac-tors to each type of ecosystem service are distinct, with natural factors generally having a larger impact than human factors. This suggests that effective management of land use and natural resources is critical for improving the supply and distribution of ecosystem services in the watershed, and should be a priority for policymakers and stakeholders working towards sustainable development goals.

In this study, five typical ecosystem services were assessed in the Kaidu-Kongque river basin, and some socio-econmic data derived formstatistical yearbooks were consult-ed to determine the accuracy of its assessment findings. However, there are still several challenges for data acquisition and coordination. These included inconsistent resolution of remote sensing data, missing attribute values of vector data, and unavailability of open key data, such as groundwater data. Furthermore, the selected time period for this study was not so extensive, and the analysis failed to capture the detailed patterns and trends of ESs and their interrelationships, as these services have spatial variability and time lag ef-fects[41-42]. In the future, it is recommended that continuous and long time series data be collected to conduct a detailed analysis of the spatial and temporal variation of different drivers and ES provision. This approach will provide a better understanding of the com-plex interactions among various factors and their influence on the ESs in the Kai-du-Kongque River basin.

We look forward to hearing from you regarding our submission. We would be glad to respond to any further questions and comments that you may have.

Round 2

Reviewer 3 Report

Dear authors,

Thank you so much to consider the suggestions. I believe that the Ms is ready for publication.

Author Response

Response to Reviewer 3 Comments

Dear Reviewers,

Thank you for your letter and the reviewers' comments on our article entitled "Spatio-temporal measurement and driving factor analysis of ecosystem service trade-offs and synergy in the Kaidu–Kongque River Basin, Xinjiang, China (ID: 2315910). It was edited in English language by MDPI to ensure correct use of grammar and common technical terms, and was edited to a level appropriate for reporting research in academic journals.

We look forward to hearing from you regarding our submission. We would be glad to respond to any further questions and comments that you may have.
